# Actin crosslinker competition and sorting drive emergent GUV size-dependent actin network architecture

Yashar Bashirzadeh [1], Steven A. Redford [2,3], Chatipat Lorpaiboon[4], Alessandro Groaz [1,10],
Hossein Moghimianavval[1], Thomas Litschel [5,11], Petra Schwille [5], Glen M. Hocky [6],
Aaron R. Dinner [2,4✉] & Allen P. Liu [1,7,8,9✉]

The proteins that make up the actin cytoskeleton can self-assemble into a variety of structures. In vitro experiments and coarse-grained simulations have shown that the actin crosslinking proteins α-actinin and fascin segregate into distinct domains in single actin bundles with a molecular size-dependent competition-based mechanism. Here, by encapsulating actin, α-actinin, and fascin in giant unilamellar vesicles (GUVs), we show that physical confinement can cause these proteins to form much more complex structures, including rings and asters at GUV peripheries and centers; the prevalence of different structures depends on GUV size. Strikingly, we found that α-actinin and fascin self-sort into separate domains in the aster structures with actin bundles whose apparent stiffness depends on the ratio of the relative concentrations of α-actinin and fascin. The observed boundary-imposed effect on protein sorting may be a general mechanism for creating emergent structures in biopolymer networks with multiple crosslinkers.

[1] Department of Mechanical Engineering, University of Michigan, Ann Arbor, MI 48109, USA. [2] James Franck Institute, University of Chicago, Chicago, IL 60637, USA. [3] The graduate program in Biophysical Sciences, University of Chicago, Chicago, IL 60637, USA. [4] Department of Chemistry, University of Chicago, Chicago, IL 60637, USA. [5] Department of Cellular and Molecular Biophysics, Max Planck Institute of Biochemistry, 82152 Martinsried, Germany. [6] Department of Chemistry, New York University, New York, NY 10003, USA. [7] Department of Biomedical Engineering, University of Michigan, Ann Arbor, MI 48109, USA. [8] Department of Biophysics, University of Michigan, Ann Arbor, MI 48109, USA. [9] Cellular and Molecular Biology Program, University of Michigan, Ann Arbor, MI 48109, USA. [10] Present address: Department of Neuroscience, Baylor College of Medicine, Houston, TX 77030, USA. [11] Present address: John A. Paulson School of Engineering and Applied Sciences, Harvard University, Cambridge, MA 02138, USA. ✉email: dinner@uchicago.edu; allenliu@umich.edu

The actin cytoskeleton endows cells with remarkable material properties[1]. They can resist significant forces but also can deform to migrate through tiny spaces. The spatial organization of the cytoskeleton is critically important for coordinating the forces that enable a cell to move, change shape, and traffic molecules intracellularly[2–4]. Actin-binding proteins cross-link filamentous actin (F-actin) into diverse network architectures, creating complex, heterogeneous materials in cells[5]. α-Actinin is a key actin cross-linker found in contractile units, particularly in actin stress fibers, which transmit traction forces across adherent cells[6]. α-Actinin dimers are about 35-nm long and bundle F-actin with a spacing that allows myosin binding and, in turn, contraction. In contrast, fascin is present predominantly at the cell leading edge in sensory protrusions such as filopodia and invadopodia[7]. There, fascin is found in tight parallel actin bundles with about 6-nm spacing[8,9]. Biomimetic motility assays with branched actin networks reconstituted on polystyrene beads formed filopodia-like bundles in the shape of aster-like patterns in the presence of fascin[10,11].

Live-cell and in vitro studies have shown that α-actinin and fascin work together in concert to enhance cell stiffness[12]. In the presence of both α-actinin and fascin, reconstituted branched actin networks on beads formed aster-like patterns and spontaneously segregated into distinct domains: α-actinin was localized to near the surface of the beads, while fascin was localized to thin protrusions[9]. Cross-linker size-dependent competitive binding effects of α-actinin and fascin can spontaneously drive their sorting and influence the association of other actin-binding proteins[9,13]. Theoretical models and coarse-grained simulations of two filaments bundling revealed that the energetic cost of bending F-actin to accommodate the different sizes of α-actinin and fascin was sufficient to drive their sorting into domains[14,15].

Although biomimetic platforms such as supported lipid bilayers[16–18] and the surfaces of giant unilamellar vesicles (GUVs)[19–21] introduce appropriate boundary conditions for self-assembly of actin-network components into biochemically and mechanically functional networks, they do not confine components in the way that a cell does. To address this issue, actin cytoskeletal components have been encapsulated within or attached to the interiors of lipid-coated single-emulsion droplets[22–25] or GUVs[25–32]. Studies that reconstitute actin and microtubule networks from purified components have revealed that spatial confinement can change the structures formed[33–36]. Here, we investigate the emergent structures and spatial organization of actin networks cross-linked by α-actinin and fascin in a spherically confining GUV environment.

## Results

### α-Actinin induces aggregation and GUV size-dependent formation of actin rings and peripheral asters.

To investigate how confinement modulates actin-network architecture, we encapsulated α-actinin together with actin inside GUVs of different sizes. We "skeletonized" z-stack confocal image sequences of actin (see "Methods" and Supplementary Figs. 1 and 2) to visualize and characterize actin bundles. Actin-bundle architecture was found to be highly dependent on GUV size but not α-actinin/actin molar ratio (Fig. 1a–b). Three types of structures were observed: single-actin rings near the GUV midplane (Supplementary Fig. 3), distinct yet connected actin bundles with no rings (networks), or a combination of connected actin bundles and actin ring(s) around the periphery (ring/network structures). In small (7–12-μm-diameter) GUVs, α-actinin-bundled actin merged into a single ring (Fig. 1c), similar to structures reported previously[36,37], with high probability. The probability of finding single rings was significantly lower in medium (12–16-μm-diameter) GUVs, and lower still in large (>16-μm-diameter) GUVs.

Increasing GUV size favored more complex actin-network structures (Fig. 1d) over rings. In particular, in the majority of large GUVs, regardless of α-actinin concentration, F-actin aggregated at the GUV periphery (Supplementary Fig. 4a, arrows), with large clusters proximal to the membrane (Fig. 1e–f, Supplementary Fig. 4b–c). We refer to these structures as "peripheral asters" below; we also describe aster-like structures with clusters in the lumen, and we refer to them as 'central asters'. We further characterized the peripheral asters by the locations of their bundles and observed that the fraction of GUVs in which all bundles were at the GUV periphery, as opposed to in the lumen, increased with α-actinin concentration (Fig. 1g). The structures formed by α-actinin–actin bundles and their dependence on GUV size are summarized schematically in Fig. 1h. Given the absence of specific binding between actin bundles and phospholipids, we interpret the preference for the periphery to result from minimization of bundle bending (i.e., elastic energy); the dependence of the bundle location on α-actinin concentration suggests that bundles span the cluster and stiffen as more cross-linkers bind. Stiffening of crossing bundles at the cluster would force actin-bundle arms out, resulting in the formation of asters.

In contrast to the results above, the cross-linker fascin formed actin bundles that were sufficiently rigid to stably deform GUVs or were sharply kinked by the membrane (Supplementary Fig. 5). Fascin–actin bundles form rings, a single-protruding bundle, or more complex structures depending on GUV size and the relative concentration of fascin to actin, consistent with previous observations[38,39]. However, fascin alone cannot induce F-actin clustering or the formation of peripheral or central actin asters.

### Encapsulated α-actinin and fascin together form distinct actin-network architectures.

The dependence of actin-network architecture on GUV size and cross-linker type motivated us to coencapsulate α-actinin and fascin together with actin in GUVs, and we found that this resulted in the formation of additional actin structures organized around clusters (Fig. 2a). These included central asters, as noted above. Actin clusters at the GUV center were always associated with an aster made up of relatively straight bundles (Fig. 2a, yellow arrow). Clusters at the GUV periphery were associated with asters as well, with bundles that curved around the GUV periphery to form partial or complete rings (Fig. 2a, white arrows). Images of such structures at early times indicated that thin actin bundles form rapidly and template further growth, which results in bundle thickening and stabilization of aster structures (Supplementary Fig. 6).

Otherwise, we observed similar trends as previously described (Fig. 2b). Figure 2c schematizes typical GUV-size-dependent actin-network architectures formed by α-actinin and fascin. The probability of actin-ring formation was reduced dramatically with increasing GUV size (Fig. 2b–d), and there was a tendency to form network (and ring/network) structures in medium and large GUVs (Fig. 2e). The majority of the network structures in the large GUVs were asters (Fig. 2f). All these trends were insensitive to the α-actinin concentration.

While the probability of aster formation was not observed to be significantly affected by α-actinin concentration, the morphology of asters was. With increasing α-actinin concentration at a fixed concentration of fascin, the cluster in the middle of a central aster grew (Fig. 3a–b). Compared with α-actinin-bundled actin structures, the addition of fascin significantly increased the probability of finding central asters (Fig. 3c). In the presence of α-actinin and fascin, increasing GUV size also increased the probability for the cluster to be localized at the GUV center (Fig. 3d). The minimization of F-actin bending energy in fascin-associated bundles together with the available space can force filaments to cross close to the center, explaining the tendency for

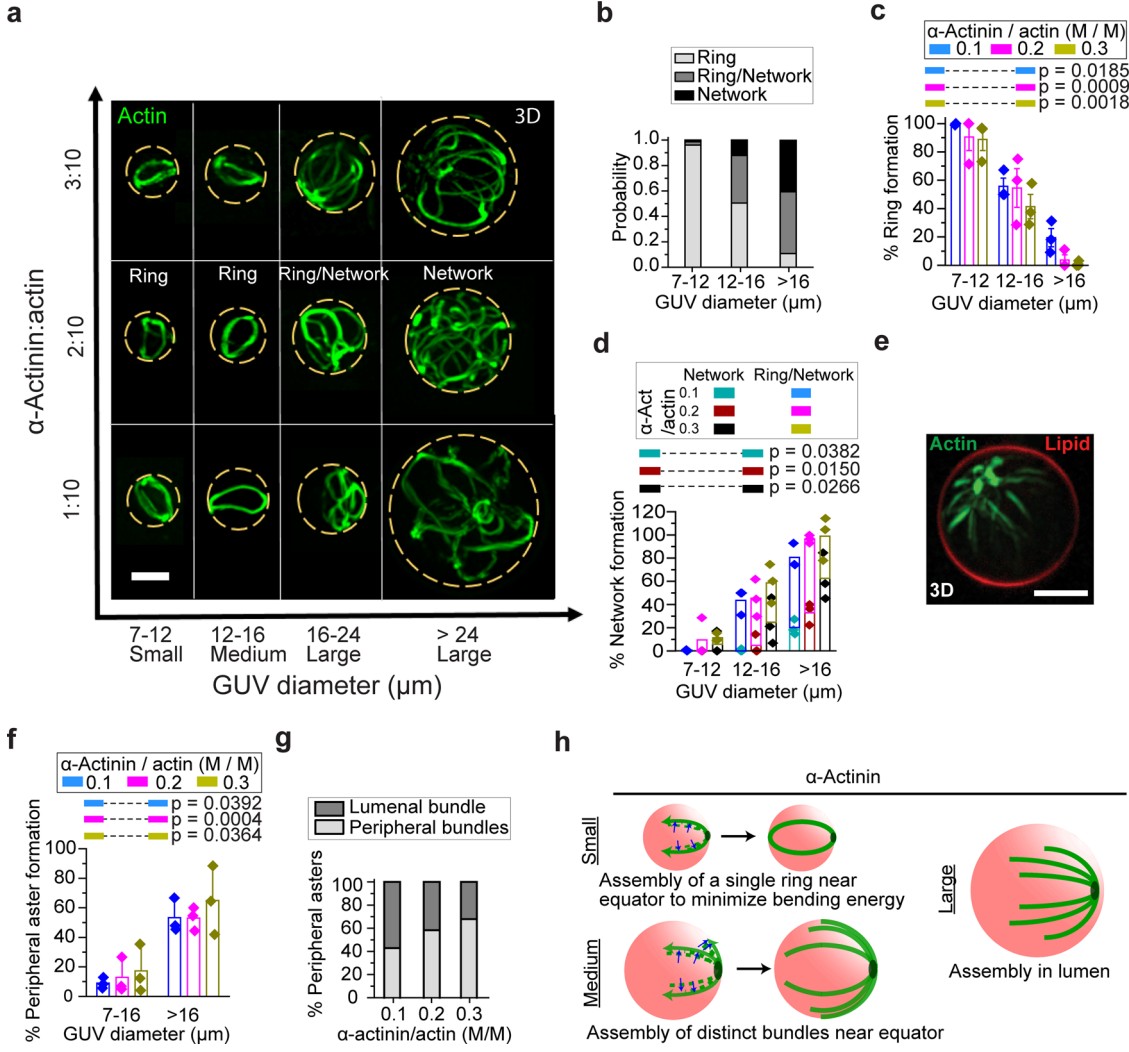

**Fig. 1 GUV size- and cross-linker concentration-dependent organization of actin-α-actinin networks. a** Representative 3D-reconstructed fluorescence confocal images of actin networks of different α-actinin:actin ratios (actin concentration at 5 μM). Actin bundles form networks in larger GUVs, while they form single rings in smaller GUVs. Dotted lines outline GUV boundaries. Scale bar, 10 μm. **b** Cumulative probability (for all three α-actinin concentrations) of ring and network formation in GUVs with different sizes. **c**, **d** Probability of the formation of rings and networks at different α-actinin concentrations. Error bars in (**d**) indicate standard error of the mean; $n = 3$ experiments. $N_{GUVs}$ per experiment = [389 42 23], [67 45 38], [188 145 128] in order of ascending α-actinin concentration (numbers in brackets are arranged in order of ascending GUV diameter). *p* values compare formation probabilities of rings (**c**) and networks (**d**) in small and large GUVs. **e** Representative 3D-reconstructed image from confocal stack of fluorescence images of an encapsulated α-actinin/actin (3:10 [M/M]) network. High α-actinin concentration can induce the formation of dense clusters at the GUV periphery. Scale bar, 10 μm. **f** The probability of the formation of actin peripheral asters depends on GUV size. The majority of actin bundles form peripheral asters in larger GUVs. Error bars indicate standard error of the mean; $n = 3$ experiments. $N_{GUVs}$ per experiment = [429 23], [112 38], [333 128] in order of ascending α-actinin concentration (numbers in brackets are arranged in order of ascending GUV diameter). *p* values compare peripheral aster-formation probabilities for the two ranges of GUV diameters. **g** Cumulative (three experiments) proportion of peripheral asters with peripheral bundles (all actin bundles elongated around GUV periphery) and luminal bundles (with at least one actin bundle elongated in GUV lumen) in large GUVs (diameter > 16 μm). At high α-actinin concentrations, the majority of actin bundles form asters with all actin bundles elongated around the periphery (peripheral bundles). Number of large GUVs with peripheral asters = [35, 14, 53] in order of ascending α-actinin concentration. **h** Schematic summary of the result of encapsulated actin-network assembly by α-actinin (without fascin) in different sized vesicles. Blue arrows show the merging of actin-filament bundles (green dashed lines) into distinct peripheral bundles (green solid lines).

clusters to be in the center of large GUVs. That the size of the central cluster but not the probability of forming a central aster changed with α-actinin concentration implied to us that fascin was driving bundling, while α-actinin was driving central clustering. We thus investigated the distribution of the two cross-linkers within asters.

**Cross-linkers spatially sort in central asters**. If fascin is indeed dominating the bundling process outside actin clusters in central

asters, the bundles' width at the same molar ratio of fascin, should be the same as the width of fascin–actin bundles in bulk, while the distribution of bundle widths with molar ratio of fascin is expected to be exponential[40]. We measured actin fluorescence intensity across fascin-actin bundles in bulk (at different molar ratios of fascin) and actin-bundle arms of encapsulated central actin asters to compare actin-bundle widths under these conditions (Supplementary Fig. 7). Intensity profiles across fascin–actin bundles in bulk showed that bundle width is larger at higher fascin molar ratios and that the intensity profiles of fascin–actin

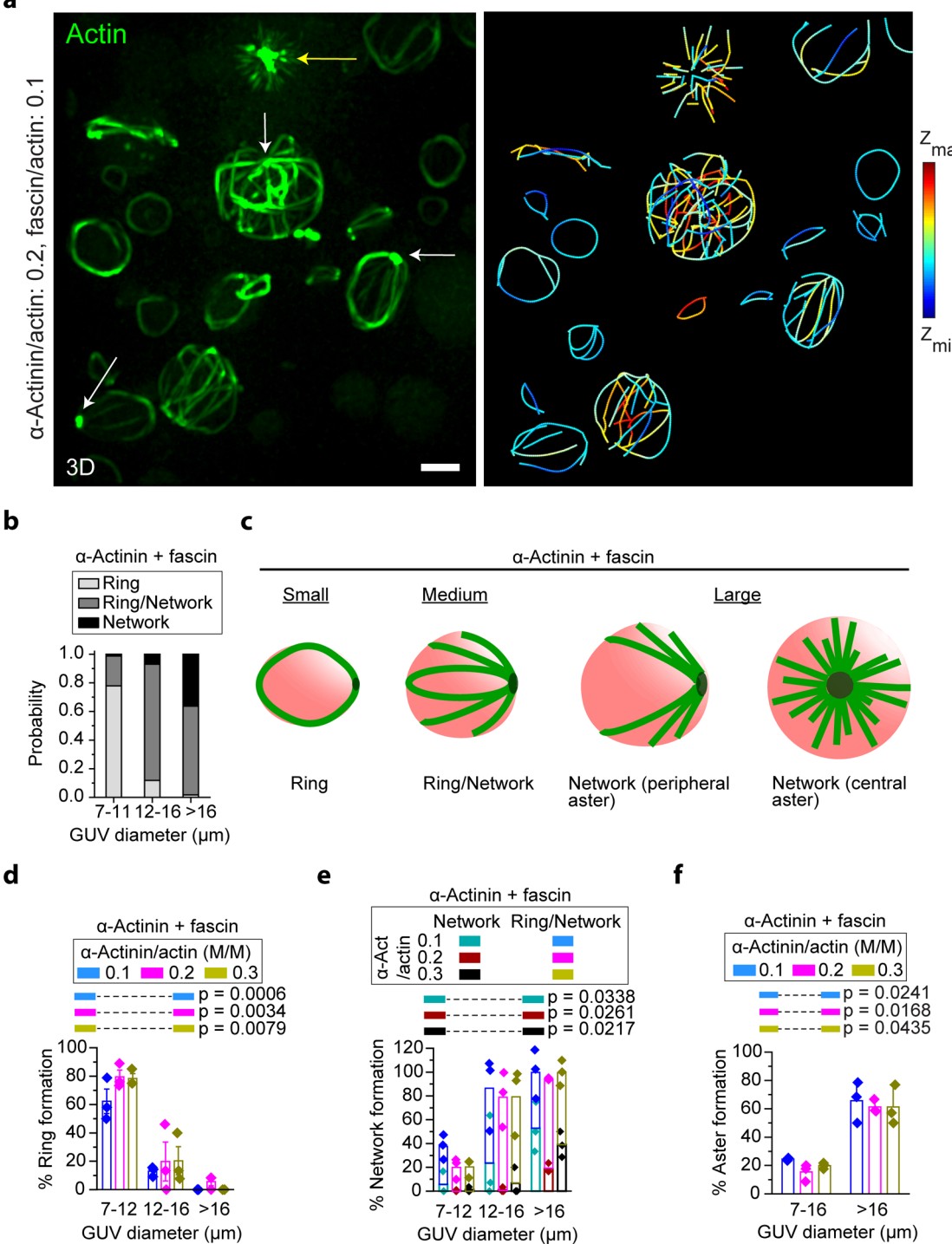

**Fig. 2 GUV-size-dependent formation of rings, peripheral asters, and central asters by α-actinin and fascin. a** Representative 3D-reconstructed (left) and skeletonized (right) images from a confocal fluorescence image stack of 5 μM actin (10% ATTO-488 actin) bundled by 0.5 μM fascin and 1 μM α-actinin in GUVs (composition: 69.9% DOPC, 30% cholesterol, and 0.1% rhodamine-PE). Yellow arrow denotes a cluster of actin fluorescence in a central aster. White arrows denote clusters of actin fluorescence in peripheral asters. Color in the skeletonized image shows z position. Scale bar, 10 μm. **b** Cumulative probability (α-actinin concentrations of 0.5, 1, and 1.5 μM with 0.5 μM fascin and 5 μM actin) of ring and network formation in GUVs with different sizes. **c** Schematic representation of GUV-size-dependent actin networks assembled by α-actinin and fascin. Rings and peripheral asters can occasionally deform GUVs. **d, e** Probability of the formation of rings (**d**) and networks (**e**) at different α-actinin concentrations as a function of GUV diameter. N$_{GUVs}$ per experiment = [101 98 29], [147 58 71], [137 82 24] in order of ascending α-actinin concentration (numbers in brackets are arranged in order of ascending GUV diameter). *p* values compare formation probabilities of rings (**d**) and networks (**e**) in small and large GUVs. **f** Probability of aster formation (peripheral or central) in the presence of fascin and α-actinin. Aster formation depends on GUV size but not α-actinin concentration. Fascin/actin, 0.1 (M/M). All error bars indicate standard error of the mean; *n* = 3 experiments. N$_{GUVs}$ per experiment = [199 29], [205 71], [219 24] in order of ascending α-actinin concentration (numbers in brackets are arranged in order of ascending GUV diameter). *p* values compare aster-formation probabilities in the two ranges of GUV diameters.

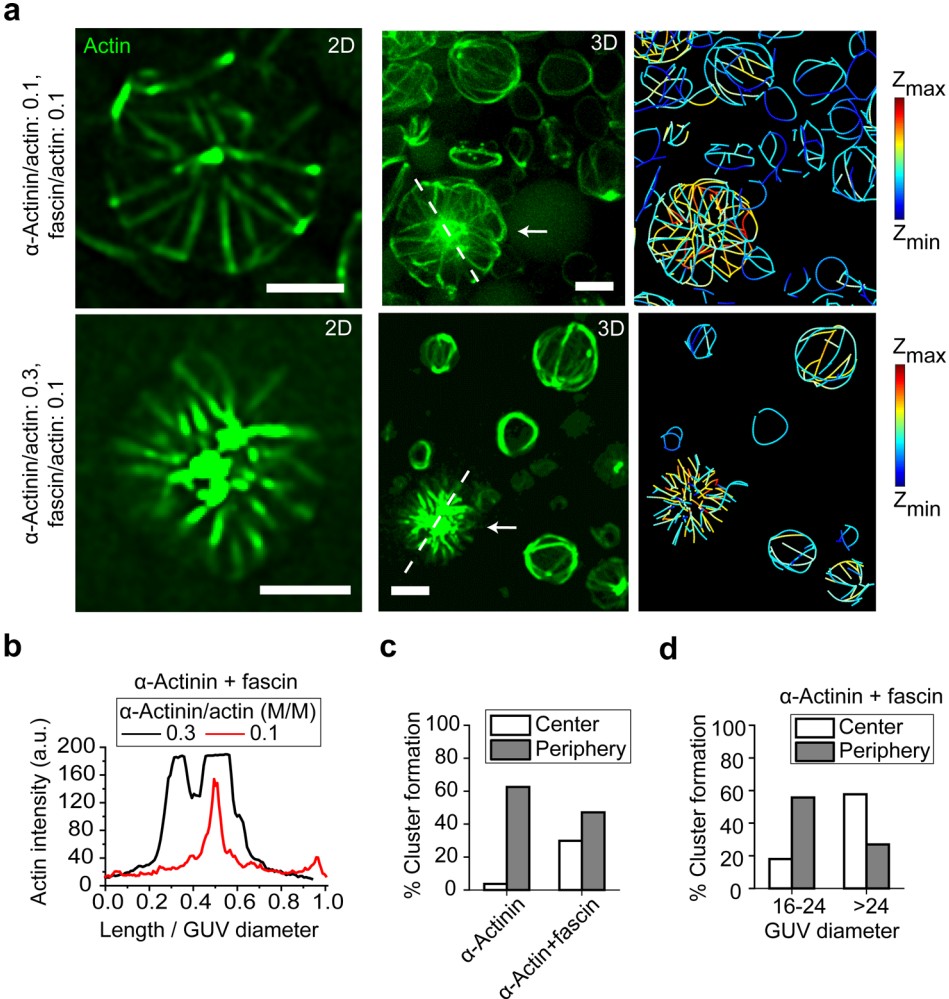

**Fig. 3 GUV-size-dependent localization of F-actin clusters in the presence of α-actinin and fascin suggests cross-linker sorting and domain formation.**
**a** Representative 2D (left) confocal fluorescence images of actin networks shown by arrows (3D-reconstructed image, middle) along with skeletonized (right) image of the GUV populations. α-actinin, 0.5 μM (top), 1.5 μM (bottom). Fascin, 0.5 μM. Actin, 5 μM. Scale bar, 10 μm. **b** Actin fluorescence intensity along the length of dashed lines drawn across the two GUVs in (**a**) normalized to the GUV diameters. **c** Cumulative probability (for all α-actinin concentrations) of the aggregation of cross-linked actin with/without fascin in GUVs with diameter >20 μm. α-actinin–fascin–actin bundles tend to shift cluster localization from the periphery to the center of GUVs. Probabilities do not add to 100% because a portion of actin bundles do not cluster at either the center or the periphery. $N_{GUVs>20\ \mu m} = 167$ [87 (α-actinin), 80 (α-actinin+fascin)]. In our analysis, only a minority of vesicles with encapsulated α-actinin–actin-bundle structures and encapsulated α-actinin–fascin–actin-bundle structures did not form clusters. **d** Cumulative probability (for all α-actinin concentrations) of the aggregation of cross-linked actin in large GUVs in the presence of both α-actinin and fascin. Larger GUVs facilitate centering of clusters. A portion of actin bundles do not cluster either at the center or at the periphery. $N_{analyzed\ GUVs>16\ \mu m} = 87$ (61 [17–24 μm], 26 [>24 μm]).

bundles formed at fascin molar ratio of 0.1 were similar to those of the actin-bundle arms of central asters formed at fascin and α-actinin molar ratios of 0.1 and 0.3, respectively (Supplementary Fig. 7d). These results suggest that fascin–actin bundles form outside the actin cluster in central aster structures.

Fluorescently labeled α-actinin was found to be localized to clusters in the middle of central asters (Fig. 4a–d, Supplementary Fig. 8a). By comparison, fascin localized throughout the entire central aster, including actin-bundle arms (Fig. 4b–d). The absence of α-actinin outside of clusters supports the hypothesis that the two crosslinkers indeed play very different roles when together (Fig. 4d, Supplementary Fig. 8a, b). Fascin dominates the region outside the clusters to form tightly packed straight actin bundles in a manner similar to that predicted for Arp2/3 complex–fascin–actin networks in bulk solution[41]. α-Actinin accumulates in the clusters and cross-links the rigid bundles, which, in the absence of significant interactions with the membrane, tend to cross at the center to minimize their bending.

In contrast to central asters, α-actinin was localized entirely along peripheral actin bundles, providing no evidence of spatial segregation (Supplementary Fig. 8a, c). α-Actinin and fascin were found to colocalize in actin rings, confirming the absence of spatial segregation along multifilament actin bundles (Supplementary Fig. 8d, e).

Observing that increasing α-actinin in the presence of fascin increased the size of actin clusters in the GUV center and enhanced the exclusion of α-actinin from actin bundles in central asters, we sought to explore the relation between this exclusion and the rigidity of the actin bundles. The persistence length of actin bundles as a function of α-actinin concentration without and with fascin showed that the bending rigidity of actin bundles is larger in the presence of both crosslinkers compared with α-actinin-bundled actin, as one would expect from the properties of the cross-linkers (Fig. 4e). Strikingly, the persistence length of α-actinin–fascin–actin bundles increased as we increased the molar ratio of α-actinin at a fixed fascin concentration (Fig. 4e). Such an

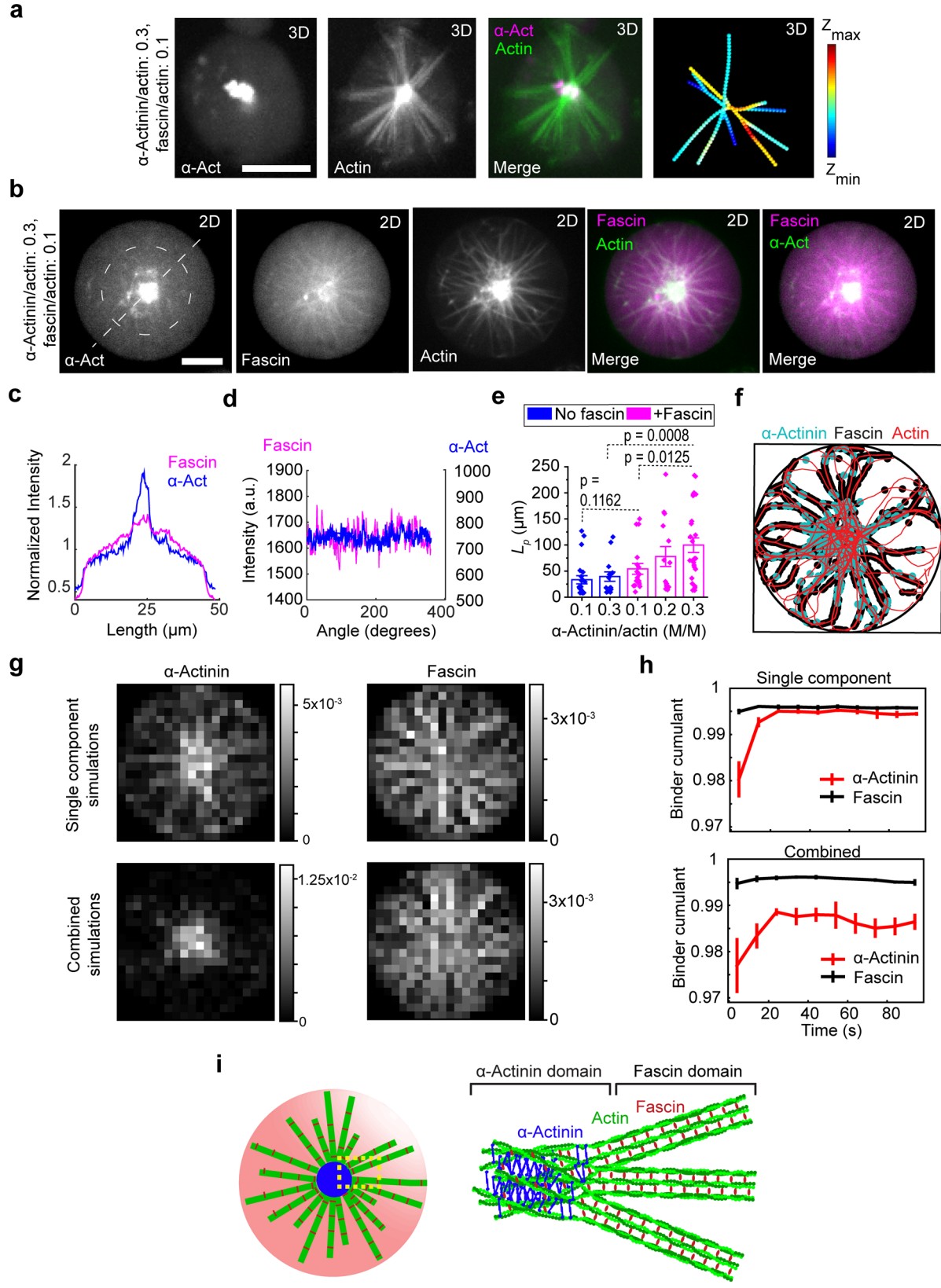

increase in bending rigidity may result from a sorting mechanism in which α-actinin stabilizes a network structure that preferentially recruits further α-actinin to the GUV center and in turn enhances bundling by fascin in the periphery. At low α-actinin concentrations, the sorting is not pronounced, and α-actinin and fascin compete to bundle actin, while, at high α-actinin

concentrations, the sorting results in tightly packed fascin–actin bundles with few α-actinin defects (Fig. 3a arrows and Fig. 4e). When α-actinin concentration was constant while varying fascin concentration, the size and density of actin clusters (Supplementary Fig. 9a, b) and bundle persistence length (Supplementary Fig. 9c) did not change. This suggests that fascin could not bind

**Fig. 4 α-Actinin and fascin sort in central aster structures. a** Representative 3D-reconstructed confocal fluorescence images of α-actinin, actin, merged, and skeletonized construct of an encapsulated central aster, respectively, from left to right. α-Actinin, 1.5 μM (including 14 mol% TMR α-actinin). Fascin, 0.5 μM (including 50 mol% AF647 Fascin). Actin, 5 μM. Scale bar, 10 μm. **b** Representative 2D confocal fluorescence images of α-actinin, fascin, actin, and merged images of an encapsulated central aster in a large GUV. α-Actinin, 1.5 μM (including 13 mol% TMR α-actinin). Fascin, 0.5 μM (including 16 mol% TMR α-actinin). Actin, 5 μM. Scale bar, 10 μm. **c**, **d** Fluorescence intensity of α-actinin and fascin along the dashed line drawn across the GUV (**c**) and circle drawn around central aster outside the actin cluster (**d**) in (**b**). **e** Persistence length of actin bundles without and with fascin (fascin/actin, 0.1 [M/M]) at different α-actinin/actin ratios indicated. $L_p$ [fascin:α-actinin:actin] = 33.3 ± 10 μm [0:1:10], 39.3 ± 9 μm [0:3:10], 54.2 ± 11.4 μm [1:1:10], 77.8 ± 19 μm [1:2:10], and 99.7 ± 15 μm [1:3:10]. $N_{bundles}$ = [22 14 17 14 26] in order of x-axis categories, 3 GUVs per category. **f** Representative structure after 100 s of simulation of actin filaments (red) cross-linked by α-actinin (cyan) and fascin (black). The border of the containing circle is shown in black. **g** PDFs of α-actinin (left) and fascin (right) from either simulation of each alone with actin (top row) or the two in combination with actin (bottom row). PDFs are constructed from the last frames of five independent simulations of 100 s for each condition. **h** Binder cumulant as a function of time for each cross-linker separately with actin (top) and the combined simulation (bottom). Values are measured for each simulation independently and then averaged. Error bars represent one standard deviation from the mean. **i** Schematic of cross-linker-size-dependent sorting in confined α-actinin–fascin cross-linked actin network.

F-actin further as fascin–actin bundles outside clusters became saturated.

To understand the microscopic origin of cross-linker sorting in GUVs, we turned to coarse-grained simulations of cytoskeletal dynamics. We used the simulation package AFINES[14,42,43] and, taking cues from previous simulations of similar systems[14], parameterized α-actinin as a relatively long and flexible crosslinker that had no preference for filament orientation and fascin as a short and stiff cross-linker that preferentially bound parallel actin filaments (see Methods). Furthermore, the two cross-linkers had different kinetics: fascin had a fast on-rate, $k_{on} = 20 s^{-1}$, whereas α-actinin had a slow on-rate, $k_{on} = 0.2 s^{-1}$. In vitro assays reveal that these two cross-linkers have similar dissociation constants when binding to actin[9], so to compensate for mechanical differences and ensure similar numbers of bound cross-linkers, the ratio $k_{on}/k_{off}$ for fascin was 40 while that for α-actinin was 4[9]. As in the experiments, fascin in the simulations produced tight, well-defined bundles, while α-actinin produced flexible bundles that were much more loosely structured (Supplementary Fig. 10). To ascertain whether these physical and kinetic differences can account for the sorting seen in experiment, we simulated a 1:1 ratio of the two cross-linkers starting from an initial condition in which actin filaments with fixed lengths crossed close to the center of the simulation region, but were otherwise randomly distributed (see Methods). After 100 s of simulation, the α-actinin (Fig. 4f, cyan) predominantly occupied the center of the aster, while the fascin (Fig. 4f, black) dominated the bundles emanating outward (Supplementary Fig. 11). To rule out possible model specificity, we ran simulations with similar parameters in the package Cytosim[44] and saw similar sorting (Supplementary Fig. 12).

To quantify these observations, we computed the probability-density functions (PDFs) of cross-linkers in the simulations. Simulations with a single cross-linker led to relatively flat PDFs (Fig. 4g, top). Simulations with the cross-linkers together led to α-actinin PDFs with a steep maximum at the center and fascin PDFs with a shallow minimum at the center (Fig. 4g, bottom). To characterize the evolution of the PDFs, we computed the Binder cumulant (see Methods) as a function of time. The Binder cumulant measures the kurtosis of an order parameter, which here is cross-linker density. A Binder cumulant value of one indicates that a distribution is close to Gaussian while a value less than one evinces a distribution with less density at extreme values on the domain considered. The value of the Binder cumulant for the fascin distribution is always high. This indicates that as fascin binds, it shows very little spatial preference, binding wherever actin is present and forming bundles that span the GUV (Fig. 4h, black curves). This is in sharp contrast to α-actinin which, both on its own and in conjunction with fascin, always exhibits a lower value of the Binder cumulant at the beginning of the simulation before relaxing to a steady value (Fig. 4h, red curves). This initially low value indicates

that, unlike fascin, α-actinin has a propensity to bind in one portion of the aster, i.e., the center, thus resulting in a tighter distribution and a lower value of the Binder cumulant, and only migrates outward into the periphery after these favorable positions have begun to be taken up (Supplementary Fig. 11). While the Binder cumulants in the single-cross-linker simulations ultimately reached similar values (Fig. 4h, top), when the cross-linkers were together, fascin was able to limit the spread of α-actinin, and a gap between the Binder cumulants persisted (Fig. 4h, bottom). These findings confirm that competition-based cross-linker sorting can give rise to the emergent structure observed experimentally.

## Discussion

This work demonstrates that confinement can have a strong effect on reconstituted actin-network architectures. We observe rings and both peripheral and central asters, depending on GUV size. As there are no motor proteins in our system, mechanisms that rely on contraction to position the structures[45–47] cannot be operative. In contrast to our system without molecular motors, the bundles' barbed ends in active myosin–fascin–actin structures face toward the aster center with myosin localized there[48–50]. A recent simulation study showed that confined actin and cross-linkers can form rings, open bundles, irregular loops, and aggregates, depending on cross-linker type and concentration, and confinement geometry[51]. Our results suggest that the positions of cross-linked actin networks can result from minimization of bundle bending as a result of membrane resistance to protrusion. Our experiments and simulations, as well as previous ones[9,14], suggest that cross-linker sorting is a factor in determining the architecture of actin-bundle networks. Although α-actinin and fascin have similar actin-bundling affinities in vitro, α-actinin has a higher binding affinity for single filaments and forms parallel and antiparallel bundles with large spacing and mixed polarity as opposed to fascin, which forms tightly packed parallel bundles[9,52]. α-Actinin can also form cross-links between bundles that are neither parallel nor antiparallel[53,54]. These properties enable α-actinin to form more complex structures even in bulk, including bundle clusters at high α-actinin concentrations[12,55,56]. In our simulated system, fascin-bundled actin filaments intersecting and coming in close proximity at the GUV center become cross-linked by α-actinin to form a central cluster (Fig. 4i). This is distinct from the convergent elongation model of how filopodia emerge from a dendritic network[57]. Similarly, the star-like actin structure with fascin-bundled actin on a bead requires nucleation of branched actin filaments[11]. Thus, our study reveals a distinct mechanism for creating an emergent actin structure from a two-cross-linker system.

In the sense that the boundary positions filaments to enable this mechanism, one can view it as driving sorting, which is distinct from spontaneous sorting in two-filament bundles[9,14]. This may be a general mechanism by which cells can select between different dynamical steady states and tune their functions as a heterogeneous

material. Boundary-driven protein sorting can be exploited in a synthetic cell context to generate more diverse self-assembling cytoskeleton structures involving multiple actin cross-linkers. Reconstitution of membrane interfaces has also revealed protein-exclusion machineries mediated by protein size and binding energy for effective localization of membrane proteins[58,59]. Thus, boundary-imposed interactions present a simple yet effective way of protein sorting in cellular processes, which induce positioning and activation of proteins at specific sites, enabling their localized functions.

## Methods

**Proteins and reagents**. Actin was either purchased from Cytoskeleton Inc, USA, or purified from rabbit skeletal muscle acetone powder (Pel-Freez Biologicals)[60]. ATTO 488 actin was purchased from Hypermol Inc, Germany. α-Actinin was purchased from Cytoskeleton Inc. Fascin was either purchased from Hypermol Inc, Germany, or purified from *E. coli* as glutathione-S-transferase (GST) fusion protein. Genes containing sequences of human fascin and an N-terminal fusion of sfGFP to fascin via a GSSG linker were cloned into pGEX-6P-1 vector (GE Healthcare) by Gibson assembly (primers listed in Supplementary Table 1). For purification, BL21(DE3) *E. coli* cells were transformed with pGEX-6P-1 fascin or pGEX-6P1-1 sfGFP-fascin. Cells were grown at 37 °C while shaking at 220 rpm, until the OD600 reached 0.5–0.6. Protein expression was induced with 0.1 mM IPTG and cell cultures were incubated at 24 °C for 8 h. Cells were harvested by centrifugation at 4000 × g for 15 min and washed with PBS once. Pellets were stored at −80 °C, until the day of purification. Cell pellets were resuspended in lysis buffer (20 mM K-HEPES, pH 7.5, 100 mM NaCl, 1 mM EDTA, and 1 mM PMSF) and ruptured by sonication. Cell lysates were centrifuged at 45,000 × g for 25 min and supernatants were loaded on a GSTrap FF 1 mL column (GE Healthcare) using an AKTA Start purification system (GE Healthcare) at a flow rate of 1 mL/min. The column was washed with 15 mL cleavage buffer (20 mM K-HEPES, pH 7.5, 500 mM NaCl). Next, 2 mL of cleavage buffer containing 160 μg of Prescission protease (TriAltus Bioscience) was loaded on the column and incubated overnight at 4 °C for cleavage. The proteins were then eluted with 5 mL of elution buffer (20 mM K-HEPES, pH 7.5, 100 mM NaCl). Purified products were dialyzed against 1 L of elution buffer twice for 3 h and once overnight at 4 °C. Protein concentration was calculated by UV absorption. Proteins were concentrated with Centricon filters (Merck–Millipore) when needed and/or diluted to a final concentration of 1 mg/mL in the elution buffer. TMR-α-actinin and Alexa Fluor 647 fascin were gifted by David Kovar (University of Chicago).

**GUV generation and microscopy**. In all, 0.4 mM stock mixture of lipids containing 69.9% 1,2-dioleoyl-sn-glycero-3-phosphocholine (DOPC), 30% cholesterol, and 0.1% 1,2-dioleoyl-sn-glycero-3-phosphoethanolamine-N-(lissamine rhodamine B sulfonyl) (Rhod-PE) in a 4:1 mixture of silicone oil and mineral oil was first made in a glass tube. The lipid/oil mixture could immediately be used or stored at 4 °C for a maximum of two days. DOPC, cholesterol, and Rhod-PE were purchased form Avanti Polar Lipids. Silicone oil and mineral oil were purchased from Sigma-Aldrich.

Next, 5 μM actin (including 10% ATO 488 actin) in polymerization buffer (50 mM KCl, 2 mM MgCl₂, 0.2 mM CaCl₂, and 4.2 mM ATP in 15 mM Tris, pH 7.5) and 5% OptiPrep was prepared and kept in ice for 10 min. α-Actinin (0.5–1.5 μM) and/or fascin (0.5–1.5 μM) were then added to the sample. GUVs were generated in 20–30 s after the addition of cross-linkers.

GUVs were generated by a modification of the cDICE method[61] (Supplementary Fig. 1a). A rotor chamber was 3D-printed with Clear resin by using a Form 3 3D printer (Formlabs) and mounted on the motor of a benchtop stir plate and rotated at 1200 rpm (60 Hz)[36]. About 0.71 mL of aqueous outer solution (200 mM D-glucose matching the osmolarity of inner solution) and around 5 mL of lipid-in-oil dispersion are sequentially transferred into the rotating chamber. The difference in density between the two solutions results in the formation of two distinct layers with a vertical water/oil interface at their boundary. To generate GUVs, a water-in-oil emulsion was first created by vigorously pipetting 15–20 μL of actin-polymerization solution (including 5% OptiPrep) in 0.7 mL of lipid/oil mixture and the emulsion was pipetted into the rotating chamber. It should be noted that the prepared mixture of actin-bundling proteins is added to the actin solution 3–4 s prior to encapsulation, and generated droplets travel through the lipid dispersion. Lipids are adsorbed at the droplet interface to form a monolayer. As the droplets cross the vertical water/oil interface, they acquire the second leaflet of the bilayer and get released in the outer solution as GUVs (bottom panel of Supplementary Fig. 1a).

The outer solution containing GUVs was transferred to a 96-well plate for microscopy. The presence of OptiPrep in GUV lumen increases GUV density and helps GUVs to sediment on the bottom of well plate. Plates were imaged 1 h after the generation of GUVs, unless otherwise mentioned. Images in Fig. 4b and Supplementary Fig. 7c were captured with a CFI APO 60X oil TIRF NA 1.49 objective on a Nikon Ti2 microscope equipped with a spinning disk confocal (Yokogawa CSU-X1), a LUNF-XL Laser launch (Nikon), and a Prime 95B CMOS 16-bit camera (Teledyne Photometrics). Image acquisition was controlled by NIS-Elements software (Nikon). All other images

were captured using an oil immersion 60 x/1.4 NA Plan-Apochromat objective on an Olympus IX-81 inverted microscope equipped with a spinning-disk confocal (Yokogawa CSU-X1), AOTF-controlled solid-state lasers (Andor Technology), and an iXON3 EMCCD camera (Andor Technology). Image acquisition was controlled by MetaMorph software (Molecular Devices). Actin- and lipid-fluorescence images were taken with 488-nm laser excitation at exposure time of 350–500 ms and 561-nm laser excitation at exposure time of 20–25 ms, respectively. A Semrock 25-nm quad-band band-pass filter was used as the emission filter. Z-stack image sequence of actin and lipids was taken with a step size of 0.5 μm. For time-lapse imaging, we used an oxygen scavenger, Oxyrase, in the encapsulated system to reduce photobleaching. It should be noted that rapid sorting and formation of aggregates occur prior to complete settlement of GUVs on the substrate despite addition of an even higher density-gradient medium (12.5% OptiPrep), in the encapsulated system. This did not allow us to capture the early formation of actin cluster by actin filaments and bundles at the center.

**Image analysis**. Image processing and image data analysis were performed using ImageJ/Fiji[62,63], SOAX[64,65], and custom MATLAB scripts (Supplementary Fig. 2). All 3D images shown are maximum projections of z-stack confocal image sequences using 3D Project command in ImageJ/Fiji. For 3D characterization of actin-bundle structures, we generated skeletonized models from regions of interest in actin images. In order to optimize the images for the identification of actin bundles, images were first preprocessed using Fiji (see Supplementary Methods). The structures from z-stack images are identified and extracted with SOAX source code[64,65] by active contour methods.

SOAX program stores all the coordinates of snakes (skeletonized bundles) and joints in a.txt file. Custom MATLAB routines were written to reconstruct the text as a Chimera marker file, include a colormap for z coordinates, and save the file as .cmm format. This process enabled UCSF Chimera[66] to read the file and provide a better 3D visualization of actin structures for selecting actin bundles and measuring parameters, such as bundle length and radius of curvature, bond vectors, ring center of mass, and persistence length with MATLAB. GUV geometrical parameters such as diameter and membrane curvature were directly measured from lipid z-stack image sequences via ImageJ.

For more details regarding image processing and data analysis see Supplementary Methods.

**Percentages and probabilities**. After taking Z-stack confocal image sequences of GUVs (561 nm) and encapsulated actin (488 nm), and image preprocessing of actin images, 3D-reconstructed actin images were obtained via ImageJ brightest-point projection using x- and y-axis, separately, as the axis of rotation. Both 3D-reconstructed and z-stack images were used for determining the number of GUVs with encapsulated actin bundles, including those with certain structural phenotype (single ring, peripheral asters, and central asters). GUV diameters were measured by line scans from both raw actin images and GUVs. GUVs were then categorized as small (7–12 μm), medium (12–16 μm), and large (>16 μm). The probability of the formation of an actin ring, ring/network, and network per GUV category per experiment was obtained by their count divided by the total number of captured GUVs in the specified category. The percentage of peripheral and central asters in large (or small/medium) GUVs per experiment was obtained by their count divided by the total number of captured large (or small/medium) GUVs with encapsulated actin bundles. The percentage of large (or small/medium) GUVs with actin clusters positioned at the center (or periphery) was calculated by dividing their count by the total number of large (or small/medium) GUVs with encapsulated actin bundles. GUVs encapsulating fluorescent actin monomers with no sign of bundling activity were occasionally found in each population. These GUVs were not taken into account for probability distribution and percentage measurements.

**Calculation of persistence length**. Using coordinates of the bonds and joints of each skeletonized actin bundle, MATLAB scripts were written to calculate orientational correlation function, $\langle C(s)\rangle \equiv \langle\cos(\theta(s))\rangle$, as a function of arc length $s$ along the contour length of selected actin bundles. $\cos(\theta(s))$ is the cosine of the angle between snake-bond vectors separated by $s$. $\langle\ \rangle$ denotes ensemble average over all snake bonds as starting points. To avoid membrane-curvature effect on persistence-length measurement, selected actin bundles were among those with no interaction or proximity to the membrane. The lengths of selected actin bundles were $8 < L < 20$ μm.

Assuming that exponential decay of $\langle C(s)\rangle$ in 3D can be described as $\langle C(s)\rangle = C_0 e^{-s/L_p'}$, we fitted lines by linear regression to data points $(s, \langle-\text{Ln}(C(s))\rangle)$ and determined the slope $-1/L_p'$ with $L_p'$ denoting the effective persistence length[67,68]. Among the selected skeletonized bundles, only those with coefficient of determination $R^2 > 0.8$ were picked for persistence-length measurement. The absolute value of the intercept, $\text{Ln}(|C_0|)$ for selected bundles was found to be around zero with a maximum of 0.03, underscoring the feasibility of assumption $C_0 = 1$ for persistence-length measurement of actin bundles with length <20 μm[69]. We did not find any correlation between persistence length and length of the selected actin bundles for any given experimental condition.

**Statistics and reproducibility**. Error bars in plots of experimental data represent the standard error of the mean. Probability and percentage measurements are based on three independent experiments. The reported *p* values are calculated by

performing unpaired two-sample student $t$-tests assuming unequal variances with confidence interval of 0.95.

**Simulation methods and analysis**. To simulate cytoskeletal networks, we used the AFINES simulation package, which is described in detail elsewhere[14,42,43] and is summarized here. It employs a coarse-grained representation of components to simulate cytoskeletal dynamics efficiently. Specifically, filaments are modeled as worm-like chains that are represented by beads connected by springs; both the bond lengths and bond angles are subject to harmonic potentials. Similarly, cross-linkers are modeled as linear springs with sites (beads) on each end that can bind and unbind from filaments via a kinetic Monte Carlo scheme that preserves detailed balance. Molecular motors can also be described within this framework, but are not included here as they are not present in the experimental system.

The positions of filaments and cross-linkers evolve by overdamped Langevin dynamics in two dimensions. Because the simulation is two-dimensional while the system of interest is not and because we expect the behavior of the experiment to be dominated by network connectivity, we neglect excluded volume interactions between unbound components. However, cross-linker heads can bind to the same filament only if they are greater than a distance $o_c$ μm apart, where $o_c$ is a parameter to control the occupancy. While this may lead to quantitative artifacts in the rate of structure formation, previous work has demonstrated that this model is effective in its description of a number of in vitro cytoskeletal systems[14,43,70,71].

In addition to the features detailed in previous work[14,42], we have added a circular confinement potential $U^{\text{confine}}$ to mimic the confinement inside the GUV, and an alignment potential $U^{\text{align}}$ between actin filaments connected by fascin or α-actinin. The potential energies of filaments ($U_f$) and of cross-linkers ($U_{xl}$) are now

$$U_f = U_f^{\text{stretch}} + U_f^{\text{bend}} + U_f^{\text{confine}} \tag{1}$$

$$U_{xl} = U_{xl}^{\text{stretch}} + U_{xl}^{\text{bind}} + U_{xl}^{\text{confine}} + U_{xl}^{\text{align}} \tag{2}$$

For brevity, we only describe the added terms. The confinement potential $U^{\text{confine}}$ is implemented as a radial confinement potential starting at radius $r_c$ with a tunable force constant $k_c$,

$$U^{\text{confine}} = \begin{cases} \frac{1}{2} k_c (r - r_c)^2, & r \geq r_c \\ 0, & r < r_c \end{cases} \tag{3}$$

where $r$ is the distance of the filament or crosslinker bead from the origin.

For α-actinin we set $U_{xl}^{\text{align}} = 0$. For fascin,

$$U_{xl}^{\text{align}} = k_{xl}^{\text{align}}(1 - \cos\theta) \tag{4}$$

where $k_{xl}^{\text{align}}$ is the penalty parameter for the angle $\theta$ between the springs of the bound filaments. This term penalizes binding to filaments that are not parallel. The penalty parameter for fascin was set to $k_{xl}^{\text{align}} = \frac{1}{3}$ pN.μm, while the penalty was not implemented for α-actinin.

We utilize these new features and the existing functionality of the AFINES package to model α-actinin and fascin. One chief difference between the two cross-linkers is their respective lengths, with fascin being almost a factor of six smaller[9]. As such, fascin is represented as short, $l = 0.1$ μm, and stiff, $\kappa = 1$ pN/μm, while α-actinin is relatively long, $l = 0.5$ μm, and soft, $\kappa = 0.1$ pN/μm. Sizes larger than the experimental structures are needed to compensate for the fact that the springs in the simulation are softer than actual proteins, as discussed previously[14]. That fascin bundles only parallel filaments and α-actinin has no preference between parallel and antiparallel filaments was captured through the $U_{xl}^{\text{align}}$ potential, as described above. The simulation parameters are summarized in Supplementary Table 2. Default values[42] were used for all parameters not listed.

The initial condition of the simulation was generated by first creating each 35-bead filament centered on the origin. After creation, the segment connecting the center to the pointed end was rotated by a random angle drawn uniformly from 0 to $2\pi$. The angle of the other half of the filament was obtained by adding an angle uniformly drawn from between $-\pi/4$ and $\pi/4$ and adding this angle to that of the first segment. Thus, each filament was assigned some random initial bend. Once the angles were assigned, the filament center was moved away from the origin by translating the bead positions in $x$ and $y$. The amount of displacement in each direction was drawn from a Gaussian distribution centered at zero with a standard deviation of 3 μm. Cross-linker initial positions were drawn with uniform probability from the box centered on the origin with dimensions $r_c \times r_c$, where $r_c$ is the radius of the confining circle, which in these simulations is 15 μm. Both cross-linkers are simulated at a density of 5 μm$^{-2}$ within the vesicle. Each simulation ran for 100 s with a time step of $2 \times 10^{-5}$ s. The updated version of AFINES, including the new features described here, as well as the full configuration files used to run these simulations and the script used to generate the initial conditions, is available at https://github.com/dinner-group/AFINES/tree/sorting_guv/crosslinker_sorting_in_GUVs. The parameters of the simulation box and actin can be found in Supplementary Table 3.

To characterize the spatial distribution of cross-linkers at a given time, we construct a list of all Cartesian coordinates of cross-linkers for which both heads are bound to filaments in that frame. From this list, we compute a two-dimensional histogram of crosslinker locations with 20 bins in each dimension. This histogram is normalized by dividing the value of each bin by the area of the bin and the total number of crosslinkers considered. In this manner, we turn our histogram into a probability-density function (PDF) that is normalized to one. The images shown in Fig. 4g are averages over the PDFs for the final frames of five independent simulations.

We also compute the Binder cumulant[72], a measure of the kurtosis of the distribution. This measure is frequently used in statistical physics to determine phase-transition points in numerical simulations, and is given by

$$U_4 = 1 - \frac{\langle s^4 \rangle}{3 \langle s^2 \rangle^2} \tag{5}$$

where $s$ is an order parameter. In our case, $s$ is the value at one point of a cross-linker PDF for an individual simulation, and $\langle \rangle$ denotes a spatial average. The lines shown in Fig. 4h are averages over five independent simulations, and the error bars indicate one standard deviation from these averages.

**Reporting summary**. Further information on research design is available in the Nature Research Reporting Summary linked to this article.

## Data availability

The experimental datasets obtained during this study are available in the "figshare" repository, https://figshare.com/articles/journal_contribution/Actin_crosslinker_competition_and_sorting_drive_emergent_GUV_size-dependent_actin_network_architecture/16337442. The data that support the findings of this study are available from the corresponding author upon reasonable request.

## Code availability

The updated version of AFINES simulation package is available at https://github.com/dinner-group/AFINES/tree/sorting_guv/crosslinker_sorting_in_GUVs.

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

## Acknowledgements

We thank Giovanni Cardone and Martin Spitaler of the MPI-B Image Facility for providing FIJI image-processing tools. We thank Morgan DeSantis and Julianna Zang (University of Michigan) for help with confocal imaging using Nikon Eclipse Ti2 confocal microscope. Fluorescently labeled α-actinin and fascin were generously provided by the Kovar lab (University of Chicago). We thank Sagardip Majumder for discussions on experimental procedures and data analysis. APL acknowledges support by a Humboldt Research Fellowship for Experienced Researchers and from the National Science Foundation (1612917, 1844132, and 1817909). GMH is supported by the National Institutes of Health grant R35GM138312. ARD acknowledges support from the National Science Foundation (DMR-2011854 and EF 1935260) and the National Institutes of Health (R35GM136381). Simulation were performed on resources from the Research Computing Center at the University of Chicago.

## Author contributions

Y.B. and A.P.L. designed the experiments. Y.B. conducted the experiments and analyzed the experimental data. A.G., H.M., Y.B., and A.P.L. prepared purified proteins. Y.B., T.L., and P.S. developed the image-analysis methods. A.R.D., G.M.H., S.A.R., and C.L. designed the modeling and simulations. S.A.R. and C.L. performed and analyzed the simulations. Y.B., A.P.L., S.A.R., G.M.H., and A.R.D. wrote the paper. All authors discussed the results and commented on the paper.

## Competing interests

The authors declare no competing interests.
