## [Peer Review File · Communications Biology]

Reviewers' comments:

Reviewer #1 (Remarks to the Author):

Review of the manuscript by Bashirzadeh et al. entitled: "Actin crosslinker competition and sorting drive emergent GUV size-dependent actin network architecture".

In this paper the authors are encapsulating actin, α -actinin, and fascin in giant unilamellar vesicles (GUVs), to study the role of physical confinement on actin network architecture. The authors show that the encapsulation of alpha actinin and fascin form rings and asters at the GUV peripheries and centers and show that the prevalence of the different structures depends on GUV size. Moreover, they show that like previous studies performed in bulk solution, α -actinin and fascin self-sort into separate domains. Among the structures formed "central" asters have actin bundles whose apparent stiffness depends on the ratio of the relative concentrations of α -actinin and fascin. The authors conclude their paper by stating that the observed boundary-imposed effect on protein sorting may be a general mechanism for creating emergent structures in biopolymer networks with multiple crosslinkers.

I find the work very interesting. The fact that one can study these systems under confined conditions is very appealing. Yet, while confinement is indeed expected to play an important role on the structure formed, sorting of alpha actinin and fascin to distinct regions of the network occurs also in bulk solution, suggesting that this is an inherent property of these two proteins rather than the confinement itself. This should be clarified along the text, especially in the discussion and conclusions. Besides these general comments, some more specific questions/remarks are given below:

Experimental work:

- How long does it take the GUVs to form in comparison to the time it takes for actin polymerization to occur?
- In the same line, does network assembly takes place, before, during, or after GUVs formation?
- Lipid composition: What is the need for such elevated amounts of Cholesterol?. Does cholesterol play any role in the GUVs formation and/or stability?
- Encapsulation: Does it affect actin encapsulation efficiency?
- What is the variance in the encapsulation efficiency of the actin solution inside the GUVs?. Any dependence on GUVs size and/or system protein composition?
- The authors state that "It should be noted that photo-bleaching of fluorophores significantly impaired actin network self-assembly at the early stages of actin bundling in GUVs. This prevented us from capturing the dynamics of self-assembly by z-stack imaging at a high-temporal resolution".

Why not use an efficient antibleaching reagent? The initial times are very important for understanding the self-assembly process, notably, the sorting of alpha actinin and fascin within the formed network.

- Alpha actinin is fluorescently labeled but not fascin. Yet, understanding how fascin distributes along the bundles and where it localizes is highly important. For instance, it is assumed that fascin localizes along rings together with alpha actinin, yet, this is not demonstrated explicitly (e.g., Fig. S6a, see also my remark below).

In addition:

In p. 3 the authors write: "There, fascin is found in tight parallel actin bundles with about 6 nm spacing."

Relevant citations should be added.

Also in p.3 the authors write: "To address this issue, actin cytoskeletal components have been encapsulated within or attached to the interiors of lipid-coated single emulsion droplets²¹⁻²³ or GUVs²³⁻²⁶."

There are other relevant papers that are missing, including those from the C. Sykes, J. Spatz, and Bausch A. groups that should be cited. There might be additionally relevant works.

P. 4 (top). Fig. S5 depicts actin fascin bundles encapsulation in GUVs of increasing size.

A similar behavior has been observed in a previous work by Claessens et al. PNAS (2008). Rings assemble in small droplets whereas more complex structures were observed in larger droplet. These results should be compared to the data presented here.

p. 5 (end): the authors write: "size of the central cluster but not the probability of forming a central aster changed with alpha actinin concentration implied to us that fascin was driving bundling while alpha-actinin was driving central clustering.

The authors do not follow the evolution of the structure with time. Yet, they could check this by comparing the distribution of the bundles' width (e.g., by measuring the fluorescence intensity of the bundles cross-section) with the distribution obtained in actin fascin bundles formed in bulk solution (see Haviv et al. EBJ 2008). If fascin is indeed dominating the bundling process a similar (exponential) distribution should be expected.

p.6 (first paragraph): Fluorescently labeled α -actinin was found to be localized to clusters in the middle of central asters (Figure 4a-b, Supplementary Fig. S6a). The absence of α -actinin outside of clusters supports the hypothesis that the two crosslinkers indeed play very different roles when together (Figure 4c, Supplementary Fig. S6a,b). α -Actinin accumulates in the clusters and crosslinks the rigid bundles, which, in the absence of significant interactions with the membrane, tend to cross at the center to minimize their bending. In contrast to central asters, α -actinin was localized entirely along peripheral actin bundles, providing no evidence of spatial segregation (Supplementary Fig.S6a, c).

The results depicted in Fig. S6a show GUVs of about the same size. In one of the GUVs a central aster is formed whereas in the other GUV a ring is formed.

1) What determines the structure formed? Could it be that in one of the GUVs there is a very small amount of encapsulated fascin (the ring case) and in the second there is a much larger amount of fascin (central aster)? Can the authors confirm that the two GUVs have the same amounts of proteins encapsulated?

In the same line, what is the variation in protein composition between the various GUVs?

2) Also, the authors state that "in contrast to central asters, α -actinin was localized entirely along peripheral actin bundles, providing no evidence of spatial segregation (Supplementary Fig.S6a, c)." This assumes that fascin is present and localizes along the ring perimeter, with no sorting effects. Yet, this is not directly demonstrated. I strongly urge the authors to label fascin.

p. 6 (end) the authors write: " At low α -actinin concentrations, the sorting is not pronounced, and α -actinin and fascin compete to bundle actin, while, at high α -actinin concentrations, the sorting results in tightly packed fascin-actin bundles with few α -actinin defects."

The authors should mention the relevant figure number and add arrows to point towards those defects.

P.8 the authors write: " Although α -actinin and fascin have similar actin bundling affinities in vitro". The authors should add relevant references that show this fact.

Also, provide relevant refs for the on and off rates (namely, binding/unbinding constants) of alpha-actinin and fascin. This remark is also relevant for the choice of parameters in the simulations where relevant citations are missing.

p.8 the authors state that: "In our system, fascin-bundled actin filaments intersecting and coming in close proximity at the GUV center become crosslinked by α -actinin to form a central cluster (Fig. 4h)."

This statement is related to simulation results, as it is not directly measured experimentally.

The authors should emphasis this in the text.

In addition. Regarding protein sorting and central aster formation: Protein sorting is not unique to passive crosslinkers and it has been shown also in systems consisting of an active crosslinker (myosin motors) and fascin. In those experiments, myosin motors were shown to sort to the center of (central) asters (Backouche et al. Phys. Biol. 2006 and Ideses et al. Soft Matter 2013) and to the junctions (local asters) of interconnected networks (Ideses et al. Nat. Comm. 2018). Like the case presented here, sorting is spontaneous. Note though that in contrast to alpha actinin and fascin system, sorting is an active process. Moreover, the bundles plus end are all facing towards the aster center, in contrast to the current system. The authors should discuss the similarities/differences of these two systems with respect to their results.

In p.8 it is written: "...fascin, which forms tightly packed parallel bundles¹¹. α -Actinin can also form crosslinks between bundles that are neither parallel nor antiparallel⁴¹.

Citation of relevant refs. describing the structural property of actin fascin and actin α actinin bundles should be added.

p.8-p.9 (last paragraph): regarding the conclusions presented in this paragraph: confinement is indeed expected to influence the structures formed. We cannot also exclude the possibility that it is playing a role in protein sorting. Yet, alpha actinin and fascin have been shown to sort spontaneously also in bulk solution (i.e., in unconfined conditions), inferring that this is an intrinsic property of these proteins. The authors should revise the text accordingly.

Image analysis:

- Calculation of Persistence Length. The authors extract the bundles persistence length L_p by calculating the orientational correlation function. To avoid membrane curvature effect on persistence length measurement only bundles with length ranging between 8-20 μm are used. According to the data presented in Fig. 4 and S7, L_p values are ranging between 35-100 μm (depending on experimental conditions). I don't see how for straight bundles whose length (8-20 μm) is much smaller than the persistence length one can obtain the decay constant, L_p , from this analysis.

Simulations:

The authors should better clarify the choice of parameters in their simulations and provide relevant support from the literature for the parameters used.

Table S2:

What is the meaning of an actin length of 0.5 μm ? I don't see how this fits the initial conditions presented in Fig. S8a.

Also in p.14 top it is written: "While this may lead to quantitative artifacts in the rate of structure formation, previous work has demonstrated that this model is effective in its description of a number of in vitro cytoskeletal systems."

Please provide relevant citations.

Reviewer #2 (Remarks to the Author):

Liu and colleagues encapsulate actin, alpha-actinin and fascin into GUVs with different sizes and analyze the polymer structures formed under different conditions.

Major findings are:

The actin bundle architecture depends on the GUV size but not on the alpha-actinin/actin molar ratio. This leads to the formation of 3 different structures, rings, actin bundles with no rings and a combination thereof.

Encapsulation of alpha-actinin and fascin leads to new architectures, actin structures organized around clusters. The structures are also GUV-size dependent. Large GUVs preferentially formed large asters. Adding fascin increased the probability to generate central asters.

The authors propose that fascin drives bundling while alpha-actinin is responsible for central clustering.

The localization of alpha-actinin and fascin is different within the clusters.

Similar structures were generated by coarse-grained simulations.

The main conclusion is that confinement has a strong effect on reconstituted actin network architectures. Furthermore, cross-linker sorting is an important factor that determines the architecture of actin bundle networks.

The presented work thus extends our view on actin bundling and proposes an important role for boundaries in driving/affecting sorting, which is of general interest to the actin field.

We thank the reviewers for their suggestions and constructive comments and have addressed them to the best of our ability. Our response is detailed below in a point-by-point manner. The original comments of the reviewers are in black text, while our responses to those comments are underlined. We have made corresponding changes in the main text and major changes are highlighted in red font in the revised manuscript.

Reviewer #1 (Remarks to the Author):

In this paper the authors are encapsulating actin, α -actinin, and fascin in giant unilamellar vesicles (GUVs), to study the role of physical confinement on actin network architecture. The authors show that the encapsulation of alpha actinin and fascin form rings and asters at the GUV peripheries and centers and show that the prevalence of the different structures depends on GUV size. Moreover, they show that like previous studies performed in bulk solution, α -actinin and fascin self-sort into separate domains. Among the structures formed “central” asters have actin bundles whose apparent stiffness depends on the ratio of the relative concentrations of α -actinin and fascin. The authors conclude their paper by stating that the observed boundary-imposed effect on protein sorting may be a general mechanism for creating emergent structures in biopolymer networks with multiple crosslinkers.

I find the work very interesting. The fact that one can study these systems under confined conditions is very appealing. Yet, while confinement is indeed expected to play an important role on the structure formed, sorting of alpha actinin and fascin to distinct regions of the network occurs also in bulk solution, suggesting that this is an inherent property of these two proteins rather than the confinement itself. This should be clarified along the text, especially in the discussion and conclusions. Besides these general comments, some more specific questions/remarks are given below:

Response: We thank the reviewer for the favorable assessment of our work. Indeed, alpha-actinin and fascin are known to sort into distinct regions on actin filaments in the bulk and around beads. However, fundamentally different structures were observed in the bulk and around beads, demonstrating that confinement modulates the structures. In particular, the same protein mixtures in bulk and around beads do not self-assemble into central asters with crosslinker sorting.

Experimental work:

1. How long does it take the GUVs to form in comparison to the time it takes for actin polymerization to occur? In the same line, does network assembly takes place, before, during, or after GUVs formation?

Response: Actin is first added into polymerization buffer on ice. Prepared mixture of actin bundling proteins is added to the actin solution 3-4 seconds before adding lipids and mixing by pipetting up and down. Droplets are formed immediately upon mixing. Thus, bundling and structure formation occurs in confinement. GUVs are generated immediately after adding them into the cDICE chamber and passing through lipid/outer solution interface. It should be noted that to avoid actin polymerization, we do not allow actin in polymerization buffer to reach room temperature prior to generation of GUVs.

2. Lipid composition: What is the need for such elevated amounts of cholesterol? Does cholesterol play any role in the GUVs formation and/or stability?

Response: We use several methods for generation of GUVs in our lab and established that 20-30 mol% cholesterol is an optimized concentration for membrane fluidity and GUV stability. We have used 30 mol% cholesterol for consistency across our experiments. Physiologically, this value is well within the range of cholesterol concentration found in plasma membranes of mammalian cells.

3. Encapsulation: Does it affect actin encapsulation efficiency? What is the variance in the encapsulation efficiency of the actin solution inside the GUVs? Any dependence on GUVs size and/or system protein composition?

Response: We characterized encapsulation efficiency and found that encapsulation efficiency is ~90% when water-in-oil emulsions are generated first and then GUVs are generated by pipetting the droplets into the rotating chamber. We have not observed a dependence on GUV size or composition. There is likely variance in terms of exact concentrations of encapsulated solutions, but they are well mixed before being dispersed into droplets.

4. The authors state that “It should be noted that photo-bleaching of fluorophores significantly impaired actin network self-assembly at the early stages of actin bundling in GUVs. This prevented us from capturing the dynamics of self-assembly by z-stack imaging at a high-temporal resolution”. Why not use an efficient antibleaching reagent? The initial times are very important for understanding the self-assembly process, notably, the sorting of alpha actinin and fascin within the formed network.

Response: This an important point. Photo-toxicity is a well-known issue in actin reconstitution field. Thanks to the referee’s suggestion, we performed additional experiments using an oxygen scavenger, Oxyrase, in the encapsulated system and could capture earlier events of actin network assembly at low exposure. We reported these results on page 5: “Earlier events of actin network self-assembly indicated that thin actin bundles form a template of actin bundle arms in aster-like structures. Actin bundling activity is dominant at the location of the template which results in thickening and formation of stable actin asters (Supplementary Figure S6)”.

It should be noted that rapid sorting and formation of aggregates occur prior to complete settlement of GUVs on the substrate despite addition of even higher density gradient medium (12.5% OptiPrep) in the encapsulated system. This did not allow us to capture the early formation of actin clusters by actin filaments and bundles at the center.

5. Alpha actinin is fluorescently labeled but not fascin. Yet, understanding how fascin distributes along the bundles and where it localizes is highly important. For instance, it is assumed that fascin localizes along rings together with alpha actinin, yet, this is not demonstrated explicitly (e.g., Fig. S6a, see also my remark below).

Response: As suggested by the reviewer, we now added labeled fascin to our encapsulated system and simultaneously imaged actin, α -actinin, and fascin in GUVs. We showed evidence

that while α -actinin-actin bundles intensely cluster at the center of central actin bundle asters, stiff actin bundle arms outside these clusters consist of fascin-actin bundles (Figure 4b-d). As our simulations predicted, fascin is indeed present in actin clusters, however, α -actinin dominates this region and as discussed previously, α -actinin is the major cause of actin network clustering in our encapsulated networks. Along this direction, we also purified GFP-tagged fascin. Supplementary Figure S5b shows GFP-tagged fascin localized to actin bundles radiating from the central cluster.

We also used labeled fascin, alpha-actinin, and actin together (Supplementary Figure S8d, e) and showed that alpha-actinin and fascin can co-localize in actin rings without any indication of sorting.

In addition:

6. In p. 3 the authors write: “There, fascin is found in tight parallel actin bundles with about 6 nm spacing.” Relevant citations should be added.

Response: We have cited relevant literature here.

7. Also in p.3 the authors write: “To address this issue, actin cytoskeletal components have been encapsulated within or attached to the interiors of lipid-coated single emulsion droplets²¹⁻²³ or GUVs²³⁻²⁶.”

There are other relevant papers that are missing, including those from the C. Sykes, J. Spatz, and Bausch A. groups that should be cited. There might be additionally relevant works.

Response: We thank the reviewer for the suggestions. Additional relevant studies, which reconstitute actin networks within or attached to the interiors of lipid-coated droplets or GUVs, have been cited to support this statement.

8. P. 4 (top). Fig. S5 depicts actin fascin bundles encapsulation in GUVs of increasing size.

A similar behavior has been observed in a previous work by Claessens et al. PNAS (2008). Rings assemble in small droplets whereas more complex structures were observed in larger droplet. These results should be compared to the data presented here.

Response: We thank the reviewer for reminding us of this relevant study. We revised the paragraph at the end of page 4 and cited relevant studies:

“Fascin-actin bundles form rings, a single protruding bundle, or more complex structures depending on GUV size and relative concentration of fascin to actin, consistent with previous observations^{38, 39}. However, fascin alone cannot induce F-actin clustering or the formation of peripheral or central actin asters.”

9. p. 5 (end): the authors write: “size of the central cluster but not the probability of forming a central aster changed with alpha actinin concentration implied to us that fascin was driving bundling while alpha-actinin was driving central clustering.

The authors do not follow the evolution of the structure with time. Yet, they could check this by comparing the distribution of the bundles' width (e.g., by measuring the fluorescence intensity of the bundles cross-section) with the distribution obtained in actin fascin bundles formed in bulk solution (see Haviv et al. EBJ 2008). If fascin is indeed dominating the bundling process a similar (exponential) distribution should be expected.

Response: We thank the author for the reference. We measured actin fluorescence intensity across fascin-actin bundles in bulk (at different molar ratios of fascin) and actin bundle arms of encapsulated central actin asters.

We added a paragraph in page 6 and added Supplementary Figure S7 to show the time evolution of the formation of central actin asters:

“If fascin is indeed dominating the bundling process outside actin clusters in central asters, the bundles' width at the same molar ratio of fascin, should be the same as the width of fascin-actin bundles in bulk while the distribution of bundle widths with molar ratio of fascin is expected to be exponential⁴⁰. We measured actin fluorescence intensity across fascin-actin bundles in bulk (at different molar ratios of fascin) and actin bundle arms of encapsulated central actin asters to compare actin bundle widths under these conditions (Supplementary Fig. S7). Intensity profiles across fascin-actin bundles in bulk showed that bundle width is larger at higher fascin molar ratios and that the intensity profiles of fascin-actin bundles formed at fascin molar ratio of 0.1 were similar to those of the actin bundle arms of central asters formed at fascin and α -actinin molar ratios of 0.1 and 0.3 respectively (Supplementary Fig. S7d). These results suggest that fascin-actin bundles form outside the actin cluster in central aster structures.”

10. p.6 (first paragraph): Fluorescently labeled α -actinin was found to be localized to clusters in the middle of central asters (Figure 4a-b, Supplementary Fig. S6a). The absence of α -actinin outside of clusters supports the hypothesis that the two crosslinkers indeed play very different roles when together (Figure 4c, Supplementary Fig. S6a,b). α -Actinin accumulates in the clusters and crosslinks the rigid bundles, which, in the absence of significant interactions with the membrane, tend to cross at the center to minimize their bending. In contrast to central asters, α -actinin was localized entirely along peripheral actin bundles, providing no evidence of spatial segregation (Supplementary Fig.S6a, c).

The results depicted in Fig. S6a show GUVs of about the same size. In one of the GUVs a central aster is formed whereas in the other GUV a ring is formed.

What determines the structure formed? Could it be that in one of the GUVs there is a very small amount of encapsulated fascin (the ring case) and in the second there is a much larger amount of fascin (central aster)? Can the authors confirm that the two GUVs have the same amounts of proteins encapsulated?

In the same line, what is the variation in protein composition between the various GUVs?

Response: We acknowledge that there can be variations in the amounts of encapsulated proteins even in GUVs with the same size. Unfortunately, there is no way for us to reliably and systematically investigate protein concentrations in two GUVs – it will always be subject to stochastic variations. Variations in encapsulation is one of the likely causes of variation in the formation of different architectures. Our statistical analysis of the data shown in Figure 2b, 2d, and 2f quantifies the frequencies of formation of rings, peripheral asters, and central asters by α -actinin-fascin-actin mixtures. It should be noted that small GUV size is indeed the prominent cause of ring formation of encapsulated actin bundles (Figure 1a and please also see PMID: 25799060).

Larger GUVs encapsulate larger amounts of proteins. To rule out the effect of potentially different actin concentrations, we repeated the experiments with a low concentration of actin (2.5 μ M) and concluded that GUV size and molar ratios of actin crosslinkers with respect to actin determines the structure of the network (image below) in a similar manner as those with higher actin concentration (5 μ M). i.e. Rings form in small GUVs (diameter, 7-11 μ m) whereas asters form in large GUVs (diameter > 12 μ m).

Figure 1. Representative 3D reconstructed image of actin from confocal fluorescence images of 2.5 μ M actin with α -actinin and fascin at α -actinin to actin ratio of 0.3 and fascin to actin ratio of 0.1 [M/M] respectively.

11. Also, the authors state that “in contrast to central asters, α -actinin was localized entirely along peripheral actin bundles, providing no evidence of spatial segregation (Supplementary Fig.S6a, c).”, This assumes that fascin is present and localizes along the ring perimeter with no sorting effects. Yet, this is not directly demonstrated. I strongly urge the authors to label fascin.

Response: We thank the reviewer for the suggestion. As mentioned earlier in the response letter, we used labeled fascin, alpha-actinin, and actin together (Supplementary Figure S8d, e) and showed that alpha-actinin and fascin can co-localize in actin rings without any indication of sorting. It should be noted that although α -actinin and fascin could segregate and form distinct domains in 2-filament actin bundles (PMID: 27666967), it is not clear if and how such segregation could intrinsically occur in actin bundle rings with multiple filaments.

12. p. 6 (end) the authors write:” At low α -actinin concentrations, the sorting is not pronounced, and α -actinin and fascin compete to bundle actin, while, at high α -actinin concentrations, the sorting results in tightly packed fascin-actin bundles with few α -actinin defects.”

The authors should mention the relevant figure number and add arrows to point towards those defects.

Response: We added a reference to the corresponding figure panels. We had previously added arrows to point out those GUVs.

13. P.8 the authors write:” Although α -actinin and fascin have similar actin bundling affinities in vitro”. The authors should add relevant references that show this fact.

Response: This statement as well as the rest of the sentence was paraphrased from reference number 8. The citation had been added at the end of the sentence.

14. Also, provide relevant refs for the on and off rates (namely, binding/unbinding constants) of alpha-actinin and fascin. This remark is also relevant for the choice of parameters in the simulations where relevant citations are missing.

Response: We have added the relevant reference in this section.

15. p.8 the authors state that: “In our system, fascin-bundled actin filaments intersecting and coming in close proximity at the GUV center become crosslinked by α -actinin to form a central cluster (Fig. 4h).” This statement is related to simulation results, as it is not directly measured experimentally. The authors should emphasize this in the text.

Response: We now write: “In our simulated system, fascin-bundled actin filaments intersecting and coming in close proximity at the GUV center become crosslinked by α -actinin to form a central cluster (Fig. 4h).”

16. In addition. Regarding protein sorting and central aster formation: Protein sorting is not unique to passive crosslinkers and it has been shown also in systems consisting of an active crosslinker (myosin motors) and fascin. In those experiments, myosin motors were shown to sort to the center of (central) asters (Backouche et al. Phys. Biol. 2006 and Ideses et al. Soft Matter 2013) and to the junctions (local asters) of interconnected networks (Ideses et al. Nat. Comm. 2018). Like the case presented here, sorting is spontaneous. Note though that in contrast to alpha actinin and fascin system, sorting is an active process. Moreover, the bundles plus end are all facing towards the aster center, in contrast to the current system. The authors should discuss the similarities/differences of these two systems with respect to their results.

Response: We thank the reviewer for suggesting this discussion. We extended our discussion in the “Outlook” section accordingly and cited the mentioned studies:

“As there are no molecular motors in our system, mechanisms that rely on contraction to position the structures⁴⁵⁻⁴⁷ cannot be operative. In contrast to our system without molecular motors, the bundles’ barbed ends in active myosin-fascin-actin structures face toward the aster center with myosin localized there⁴⁸⁻⁵⁰.”

17. In p.8 it is written: “...fascin, which forms tightly packed parallel bundles¹¹. α -Actinin can also form crosslinks between bundles that are neither parallel nor antiparallel⁴¹. Citation of relevant refs. describing the structural property of actin fascin and actin alpha actinin bundles should be added.

Response: We thank the reviewer for the suggestion. Additional relevant literatures have been cited in these statements.

18. p.8-p.9 (last paragraph): regarding the conclusions presented in this paragraph: confinement is indeed expected to influence the structures formed. We cannot also exclude the possibility that it is playing a role in protein sorting. Yet, alpha actinin and fascin have been shown to sort spontaneously also in bulk solution (i.e., in unconfined conditions), inferring that this is an intrinsic property of these proteins. The authors should revise the text accordingly.

Response: We thank the reviewer for the suggestion. We revised the first sentence of the paragraph accordingly and added references: “In the sense that the boundary positions filaments to enable this mechanism, one can view it as driving sorting which is distinct from spontaneous sorting in two-filament bundles^{8, 14}.”

19. Image analysis: Calculation of Persistence Length. The authors extract the bundles persistence length L_p by calculating the orientational correlation function. To avoid membrane curvature effect on persistence length measurement only bundles with length ranging between 8-20 μm are used. According to the data presented in Fig. 4 and S7, L_p values are ranging between 35-100 μm (depending on experimental conditions). I don’t see how for straight bundles whose length (8-20 μm) is much smaller than the persistence length one can obtain the decay constant, L_p , from this analysis.

Response: The persistence length was measured using the exponential decay of orientational correlation along the contour of actin bundle. Decay constant was extracted by curve fitting to the decay function as described in the Methods section, “Calculation of Persistence Length”. We acknowledge that for straight bundles where persistence length of bundles is larger than bundle length, persistence length cannot be precisely estimated. However, since we clearly detect the decay and finite decay constant via linear regression, we expect that the error arising from shorter bundle length is minor (also see reference 68. Importantly, since the difference between persistence length of fascin-actin and α -actinin-actin bundles is significant we do not expect this to have an influence on our interpretation of the results presented.

Simulations:

The authors should better clarify the choice of parameters in their simulations and provide relevant support from the literature for the parameters used.

Response: In page 7, we have added references to better justify our parameter choices and clarified the language in this section.

Table S2:

What is the meaning of an actin length of 0.5 μ m? I don't see how this fits the initial conditions presented in Fig. S8a.

Response: Response: We apologize to the reviewer for the confusion and appreciate their close reading of the manuscript. The 'actin length' is the radius of a single bead within the filament. This parameter is used to calculate the friction on a single monomer for the molecular dynamics integration. We use the default value for the software and have removed this row from Table S3 to avoid further confusion.

Also in p.14 top it is written: "While this may lead to quantitative artifacts in the rate of structure formation, previous work has demonstrated that this model is effective in its description of a number of in vitro cytoskeletal systems."

Please provide relevant citations.

Response: We have added references in this section.

Reviewer #2 (Remarks to the Author):

Liu and colleagues encapsulate actin, alpha-actinin and fascin into GUVs with different sizes and analyze the polymer structures formed under different conditions.

Major findings are:

The actin bundle architecture depends on the GUV size but not on the alpha-actinin/actin molar ratio. This leads to the formation of 3 different structures, rings, actin bundles with no rings and a combination thereof.

Encapsulation of alpha-actinin and fascin leads to new architectures, actin structures organized around clusters. The structures are also GUV-size dependent. Large GUVs preferentially formed large asters. Adding fascin increased the probability to generate central asters.

The authors propose that fascin drives bundling while alpha-actinin is responsible for central clustering.

The localization of alpha-actinin and fascin is different within the clusters.

Similar structures were generated by coarse-grained simulations.

The main conclusion is that confinement has a strong effect on reconstituted actin network architectures. Furthermore, cross-linker sorting is an important factor that determines the architecture of actin bundle networks.

The presented work thus extends our view on actin bundling and proposes an important role for boundaries in driving/affecting sorting, which is of general interest to the actin field.

Response: The reviewer provided a great summary of our major findings and had no critiques. We thank the reviewer for the positive feedback.

REVIEWERS' COMMENTS:

Reviewer #1 (Remarks to the Author):

The authors have addressed my multiple questions/comments and performed additional experiments, as I proposed.

The response regarding the measurement of the persistence length is still not fully convincing to me. But I will not insist on that. The rest is ok.